# Data Analyses of Quarry Operations and Maintenance Schedules: A Production Optimization Study

Brennan George *[ID] and Bahareh Nojabaei

Department of Mining and Minerals Engineering, Virginia Polytechnic Institute and State University, Blacksburg, VA 24060, USA; baharehn@vt.edu
* Correspondence: bkg00@vt.edu

**Abstract:** In this research, data analytics and machine learning were used to identify the performance metrics of loaders and haul trucks during mining operations. We used real-time collected data from loaders and haul trucks operating in multiple quarries to broaden the scope of the study and remove bias. Our model indicates relationships between multiple variables and their impacts on production in an operation. Data analysis was also applied to ground engagement tools (GET) to identify key preventative maintenance schedules to minimize production impact from capital equipment downtime. Through analysis of the loader's data, it was found there is an efficient cycle time of around 35 s to 40 s, which yielded a higher payload. The decision tree classifier algorithm created a model that was 87.99% accurate in estimating the performance of a loader based on a full analysis of the data. Based on the distribution of production variables across each type of loader performing in a similar work environment, the Caterpillar 992K and 990K were the highest-yielding machines. Production efficiency was compared before and after maintenance periods of ground engaging tools on loader buckets. With the use of maintenance and production records for these tools, it was concluded that there was no distinguishable change in average production and percentage change in production value before and after maintenance days.

**Keywords:** mining industry; quarry optimization; loaders; data analytics; machine learning





## 1. Introduction

The development of new technologies has opened a broader view into machines and sensors through the expanded collection of big data. The Internet of Things (IoT) has taken hold in the mining industry, allowing connections between equipment and software to produce real-time data on numerous operation parts: hazard analysis, fleet management, condition monitoring, alarm systems, and even process optimization [1–3]. Machine companies that design machines for mining purposes, such as Caterpillar, are now fitting newer generations of machines with hardware to allow big data collection [4,5]. The new hardware connects sensors across the machine to user interfaces and maintains records of all metrics. The new sensor–user interface connection then uses the structured data to output analyses of performance metrics, efficiency, and overall machine health [6].

Although these technological improvements exist, they are not always being fully utilized by companies in the mining industry. Mining is a mature industry that is accustomed to how things have been done in the past because it is cheaper than adopting new technologies in an ever-growing industry [1,7]. Technology can benefit the mining industry by providing predictive analyses so that they can be proactive and not reactive to hazards such as mine disasters or machines breaking down which delay operations. Being proactive can help predict poor working conditions to improve the health of the workers, predict the failure of machines due to parts breaking or fluid leaks causing structural damage, or even prevent deaths such as the predicted landslide at the Bingham Mine in 2013 [8,9]. Aside from predictive analyses, the new technology can also help by determining the

optimal performance of machines, providing operators with a production goal allowing the operation to achieve its full potential to keep up with demand.

The purpose of this exploratory study is to investigate the benefits of big data collection and to facilitate the identification of key performance metrics using two Caterpillar software tools, Cat Productivity (version 2.0, Caterpillar Inc., Irving, TX, USA) and Cat MineStar Edge (version 1.5.20220804p2, Caterpillar Inc., Irving, TX, USA). The two pieces of software are newer to the industry and are slowly being adopted. An analysis of the production data output from the machines would be beneficial in determining optimal production metrics to set goals for operations as well as visualizing maintenance benefits. In this study, the collected data will be utilized to identify these optimal performance metrics with the goal to increase yearly production. One hypothesis tested was to see if there was an optimum point at lower cycle times, ideally in the 30 s to 45 s range, in which the average payload will have a higher yield, thus setting a production goal for operators. Another hypothesis evaluated whether the maintenance activity on GET loader buckets is appropriate as operators currently determine the frequency of replacement. The last hypothesis states that machine learning will yield a predictive model of loader type and production values associated with that loader. Regression models can be used to find additional variable relationships not already highlighted.

This research paper is set up to provide previous studies related to the work carried out. Then, data are introduced as well as any preprocessing performed on the data before analysis. Next, all the methods are discussed with their results and a discussion of these results is detailed as well. Finally, the overall conclusions and future work are discussed.

## 2. Previous Studies

Previous studies have explored the optimization of fleet interactions without the use of technology that highly relies on equations that provide a number based on variables gathered from visual guesses as well as manual measurements. Matsimbe (2020) performed data analysis on shovel-truck interactions in a quarry in Malawi to determine if they could optimize the fleet size using different numbers of haul trucks for one shovel [10]. Using a stopwatch for the cycle time as well as numerous equations for the payload size that rely on the user's judgment of how full the shovel bucket is, they were able to determine the increased fleet size with the current size shovel, resulting in the queuing time of the haul trucks increasing by 6.40 min [10]. Nday et al. performed similar work in a mine in the Democratic Republic of Congo, with hand calculations that considered the conditions of haul roads and equipment as well as operator experience to lower cycle times by about 8% [11]. Finally, Samatemba et al. recently employed equations to discover the utilization rate, production rate, equipment availability, efficiency, and performance rate for haul trucks, loaders, and drill rigs in Chibuluma South Mine in Zambia [12]. With simple analysis, these researchers were able to determine that all their machines were working with less than a 50% effectiveness rate, translating to a large loss of revenue [12]. Imagine if their machines related sensors to software that could accurately measure the payload, the loader and hauler unit cycle times, and the trucks' queuing time. Without using equations that include user opinion, which risks potential user error, they could provide a more accurate representation of how their fleet interacts with different sized shovels and fleets. The common factor of the three mines is that they are in third-world countries without wide means to access the technology to connect their machines to software. In the future, technology access will become more cost-effective and readily available for developing countries, thus yielding a potential evolution of adoption over time.

The Inclusion of sensors and data in machines allows predictive maintenance strategies to be used more frequently. In the past, without access to technology to give information on performance metrics and machine health, maintenance would occur when machines broke down or when operators used visual or auditory cues to determine that damage had already occurred [13]. The addition of sensors and algorithms to machines can help circumvent this reactive maintenance planning. The equipment is more reliable, meaning

less downtime when parts break, and there is a noticeable cost reduction since fewer pieces of equipment are being repaired [14]. Algorithms such as those in machine learning can be utilized to create predictive models from past data to forecast and schedule maintenance for machines. Basri et al. conducted a review of this and found that through the computer-based approach to predicting machine failures, companies achieved better performance and productivity over time compared to reactive maintenance practices [15]. Due to current supply chain disruptions, it is vital for operations to predict when machines need parts instead of waiting until failure and potentially losing a machine for multiple months while waiting on replacement parts.

Machine learning has been a popular topic in numerous research fields and recent developments in technology allow for easy access and adoption. Researchers in the mining industry have focused on this trend and are producing ever-growing research to see the capabilities of these algorithms alongside the collection of big data. Nobahar et al. used five algorithms: linear regression, decision tree, K-nearest neighbors, random forest, and gradient boosting to simulate operations to optimize fleet selection [16]. The study found that the gradient boosting regressor algorithm was accurately able to predict the best fleet selection when given the performance metrics, weather conditions, and haul road routes with an 85% accuracy rate. Baek et al. utilized a deep neural network algorithm to predict the ore production of a mine in the Republic of Korea based on the performance metrics of a fleet of haul trucks [17]. The machine learning algorithm was used on two sets of data, one for morning production and one for afternoon production. The results of the deep neural network algorithm were promising as their mean absolute percentage error for morning production was 11.40% and for afternoon production, it was 8.87% [17]. A mean absolute percentage error of less than 10 is excellent, while between 10 to 25 is low but acceptable [18]. The mean absolute error compares the forecasted result to the known value, and a percentage error is calculated based on their difference [18]. Machine learning algorithms and results should continue to improve over time, and with the addition of big data collection in the mining industry, they will expectantly be used for predicting performance metrics and required maintenance.

## 3. Methodology

### 3.1. Data Collection

This study used data from two different pieces of software: CAT Productivity and CAT MineStar Edge. This software is proprietary to Caterpillar and requires individual companies to pay a subscription to have their machines connected to the network to access the data collected by sensors on the machine. These machines are connected to the network using a 4G signal transferred through what is called the Product Link box. This Product Link box allows for the health, utilization, production insights, and hours/location to be transferred to the software dashboards for use by the customer or owner of the software [19–21]. In this study, all machines used have a Product Link box of generation PLE641 which allows for the advanced production metrics to be gathered. CAT Productivity had multiple loaders, aside from the ones used in this study, that had PLE641 boxes but because the customers had not subscribed to the software, these machines only presented utilization data as well as cycle time metrics. These two pieces of software can present similar performance metrics, machine health, and utilization but there are a few key differences. CAT MineStar Edge allows for real-time data collection and playback recordings of what work machines are doing and easily ties which hauler unit is being loaded by the front-end loader [22]. On the other hand, CAT Productivity does not provide real-time data collection and does not provide as in-depth insights as CAT MineStar Edge since it is a cheaper subscription product [23]. These data are not publicly accessible, and they were provided by the company Carter Machinery Company Inc. for use in this study. All data collected in this exploratory study were scrubbed of any customer-identifying information as well as asset numbers to prevent insight into customer production information.

### 3.1.1. Cat Productivity Data Set

The first data collected from CAT Productivity consisted of variables for basic production information with a data set for each of the seven loaders. The first data group used eight months of data from April to December 2022. The loaders used in this study were in different quarries across Virginia with similar production conditions to each other. These rock quarries, across Virginia, are surface operations that produce limestone, sand, and gravel. The first group of data was collected using the software Cat Productivity. The Caterpillar loader machines were of slightly different sizes based on their generation. The seven Caterpillar loaders consisted of one 992K machine, two 990K machines, three 988K machines, and one 988K XE machine. These machines are variable in size but are frequently used for similar-size quarries based on company choice. The loader data were tied to each haul truck, thus enabling the gathering of truck metrics as well. The data collected variables for date and time, bucket payload (tons), truck total buckets and truck total payload (tons), cycle time (seconds), and truck ID. Bucket payload is just the tonnage of the material that is in the loader bucket before it is dumped into the haul truck. Truck total buckets is the total number of loader buckets full of material that it takes to fill up the truck before it departs the loading area. Truck total payload is the total tonnage of the material that the truck departs the loading area with after being filled by the loader. The cycle time is in regard to the loader, and it is measured in the total time of the following four phases: picking up material from the working face to fill the loader bucket, swinging toward the haul truck, dumping the material in the haul truck bed, and finally swinging back toward the working face [24]. Truck ID is simply a way for the companies to distinguish which truck is being loaded by the loader. Table 1 shows a sample of the data collected.

**Table 1.** Sample Cat Productivity production data for loaders from rock quarries.

| Bucket Payload Measurement Time | Bucket Payload (tons) | Truck Total Buckets | Truck Total Payload (tons) | Cycle Time (seconds) | Truck ID |
|---|---|---|---|---|---|
| 16 April 2022 | 16.94 | 4 | 68.93 | 45 | 1 |
| 12 May 2022 | 14.56 | 5 | 87.64 | 73 | 1 |

### 3.1.2. CAT MineStar Edge Data Set

The second data set consisting of the two loaders from CAT MineStar Edge had numerous variables, but only a few can be utilized for performance metrics analysis. This is because most of the information in this data set is made up of information sensitive to the company such as load and dump location in X, Y, and Z as well as latitude and longitude coordinates, haul routes with distinguishing information, and machine serial numbers. Alongside the sensitive information, there was miscellaneous information that was omitted due to it not being related to any performance metrics. For example, the variable cycle type was omitted because every data point had the same value which was "HAUL" as well as other variables such as haul operator which was not filled out for any data point. The second group consisted of two loader machines of the generation Caterpillar 993K. This second data set also consisted of eight months of data from a large-scale surface coal mine located in West Virginia. These Caterpillar 993Ks are large pieces of machinery that are capable of large production operations such as the surface coal mine this study was conducted on. The most useful variables in the data set were date, load duration, truck cycle time (seconds), plan distance full (m), loader dipper count, and resolved payload (tons). Load duration is the length of time it took for these loaders to fill the haul trucks with the material. Truck cycle time is the total time that the haul truck goes through the phases of the start of loading, hauling to the dump site, dumping, hauling back to the loading site, and then pulling up next to the loader again [25]. Plan distance full is the total horizontal distance that the haul truck travels to reach the dump site from the loading site. Loader dipper count is the number of loader buckets it takes to fill the bed of the haul truck

before it departs for the dump site [26]. Resolved payload is another name for total truck payload. Table 2 shows a sample of the data.

**Table 2.** Sample Cat MineStar Edge production data for loaders from the surface coal mine.

| Load Duration | Truck Cycle Time (seconds) | Plan Distance Full (m) | Loader Dipper Count | Resolved Payload (tons) |
|---|---|---|---|---|
| 16 April 2022 | 16.94 | 4 | 68.93 | 45 |
| 12 May 2022 | 14.56 | 5 | 87.64 | 73 |

*3.2. Data Preprocessing*

Each data set downloaded from every loader was cleaned for data analysis to be conducted and to be input into the models. This process included identifying the range of cycle times for each loader (so they were neither too large nor too small, indicating tasks other than loading haul trucks), deleting missing data, and filtering out unnecessary variables that would not indicate production metrics. Given the lack of filled-out data for some production variables, they were deleted from the data sets because of inconsistencies in subsequent data collection across each loader. For example, data that did not indicate production metrics were information such as coordinates of load and dump sites or names of haul routes identifiable to the company.

For the first data set from Cat Productivity, numerous data points needed to be deleted. Machines are hooked up to the software through devices called Product Link boxes. Product Link boxes have multiple generations that gather varying amounts of data based on the generation employed. For example, many of the loaders used the generation 541 Product Link box while two loaders used the 641 Product Link boxes. Given the mixed Product Link box generations among loaders, some variables were deleted from the data set because of inconsistencies in subsequent data collection across each loader. These variables included bucket payload sequence, truck total buckets, truck total payload, and hauling unit. The loaders with the 641 Product Link boxes were set up to identify how many bucket cycles it took to load the haul truck, the overall tonnage of the haul truck when full, and the haul truck associated with that cycle.

For the Cat MineStar Edge data set, the only variables that were deleted were associated with the loading area, the coordinates of the haul route, and the slope of the haul route. Each loader had its cycle time analyzed in a distribution to determine the frequency of task achievement, creating low and high bounds for a cut-off time. Loader operators may perform multiple tasks while waiting for haul trucks to position themselves next to the loader at the mining face. On the low end, they may be cleaning the face or floor by picking up and dumping material repeatedly in a quick fashion. On the high end, they could be sitting with a bucket full of material waiting for a haul truck to pull up.

*3.3. Data Analysis*

After the collection of the eight months of data for each loader from CAT Productivity and CAT MineStar Edge, data cleaning was conducted. Microsoft Excel was used to clean and prepare the data sets to verify the accuracy and reliability of the data. The data cleaning process involved removing sensitive information from the companies and variables that consistently had a lack of information. Following the removal of data, the fill factor for the CAT Productivity set of loaders was calculated which was conducted by dividing each bucket payload by the maximum payload possible for that machine. Due to a lack of individual cycle bucket payloads in the CAT MineStar Edge data set, the fill factor was not able to be calculated for the two loaders in the surface coal mine. Following the data cleaning process, the PivotTable function within Microsoft Excel was used to calculate the average payload for each cycle time. To determine the most common cycle times for each loader, the frequency at which each cycle time appears was calculated in each data set. This provides insights into the payload distribution and to help identify any patterns or trends

in the data. Next, the average payload and frequency of the cycle times was imported into the software Google Colab (version 3.9, Google, Mountain View, CA, USA). Utilizing this software with Python libraries, Matplotlib (version 3.7.1), Pandas (version 1.5.3), and Numpy (version 1.22.4), scatter plots were created to display the distribution of data for the average payload vs. the cycle time, and histograms were created to visualize the frequency of the cycle time for each of the loaders. These figures allow for insight into the relationships between the variables and to identify any outliers in the data.

### 3.4. Statistical Analysis

Google Colab was used to perform statistical analysis, calculations, and data visualization for all data sets. The first step was to calculate the measures of central tendency for the chosen major production metrics for each set of loaders. The measures of central tendency included the mean, standard deviation, and first and third quartile. For the CAT Productivity loaders, the seven loader units in this set were compared based on the variables of loader cycle time, fill factor, and bucket payload. For the CAT MineStar Edge loaders, the two loader units in this data set were compared based on the variables of total truck payload, loader cycle time, and truck cycle time. Through calculating the measures of central tendency, a visualization of the distribution of data as well as any trends or patterns that may appear were obtained. These can provide insights into the performance of each loader compared to each other as well as show any areas of improvement. The probability density function (PDF) is calculated using these measures of central tendency in order to visualize the distribution of the data. Through visualization of the PDF across each loader for each variable, any significant differences between the performance of the loaders in each set can be identified to gain a better understanding of the underlying data. There is a lack of significant production variables to compare across the data, so this is a limitation. Necessary judgments were made as to which variables to focus on for the comparison of the production metrics.

### 3.5. GET Maintenance Applications

The GET maintenance analysis was conducted using Microsoft Excel and Google Colab. Excel was primarily used for collecting average values of production and for cost analysis. Two data sets were initially combined, the data set of production metrics for the 988K loader that maintenance occurred on and the data set of maintenance days for replacing the GET life. Initially, the average payload value for each day was calculated using the PivotTable function. These data for the average payload across each day were then matched up with the days that maintenance was performed. The average value for payload was then calculated for a period of five days before and after maintenance, four days before and after, three days before and after, two days before and after, one day before and after, and on the day of maintenance. This gave a table of 11 values that showed the average production before and after every day of maintenance. Google Colab was then used with the matplotlib.pyplot library in Python to create a visual of the distribution of these average payload values around maintenance. Then, a cost analysis was conducted of the production value of these periods. This was achieved using some assumptions that are listed in Table 3. A price of USD 5 per ton of product was chosen for a speculative analysis of limestone rock at the time.

**Table 3.** Assumed values for calculation of production value for each set of production days leading up to and after maintenance.

| Average Price of Product (USD/ton) | Average Maintenance Time (hour) | Production Time Per Day (hours) | Average Number of Buckets Per Truck | Average Cycle Time (seconds) | Price of Teeth (USD) |
|---|---|---|---|---|---|
| 5 | 1 | 7.5 | 7 | 50 | 2181.76 |

Once the production values for each period before and after maintenance were calculated, a percentage change was then calculated by comparing the before and after values of each maintenance day. These percentage changes and maintenance days were then input into Google Colab, and again, with the matplotlib.pyplot library, they were plotted for data visualization.

### 3.6. Machine Learning

The machine learning set-up and analysis were conducted using Google Colab since it has access to the machine learning libraries within Python. Initially, all of the loader's data were combined into the same data set. Using Google Colab, they were initially put into a linear regression algorithm to determine correlations between the data in the data set. R-squared is used as an indication of model performance with these regression models. Due to the poor results, another linear regression model was created using only a random fraction of the data set to test if there were clearer results. Since linear regression produced poor results, polynomial regression was then chosen to visualize the correlation between values in the data. The data were pushed through this polynomial regression model, and the R-squared was produced. Once this was completed, a heat map was created to visualize the correlations between the variables in the data. The next step was to create models for the machine learning prediction algorithms. The four models used in this analysis were the prediction algorithms based on K-nearest neighbors, polynomial regression, decision tree, and random forest. The data input into these models were based on training and testing sets. Approximately 20% of the data were chosen as the testing set while 80% were used as the training data set. The accuracy score is the output from these models which indicates model performance.

## 4. Results and Discussion

### 4.1. Data Analysis

Through data analysis of each loader's data set, a distribution of the average payload for each cycle time was obtained. These data were split to determine if there is an optimum point for each loader at which the average payload per bucket can be increased at faster cycle times to yield higher production down the line. The frequency of each cycle time was also included to determine the spread of the data and how often the operators are achieving each cycle time. Each loader was analyzed between cycle times that were determinants of its distribution of data. The cycle time of the seven loaders that operate in rock quarries is approximately between 25 s to 80 s. Figures 1–7 show the scatterplots of the average payload vs. cycle time and the histograms of the frequency of each cycle time.

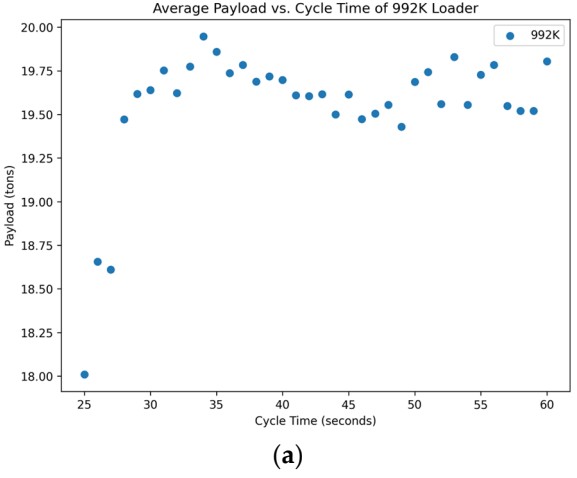

(**a**)

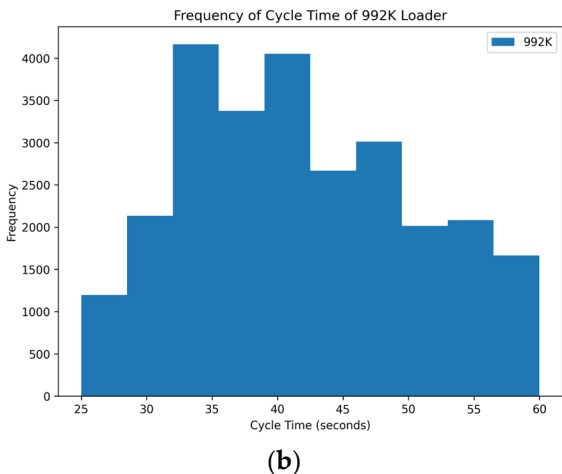

(**b**)

**Figure 1.** Distribution of variables within the 992K data set: (**a**) scatterplot of average payload per bucket and cycle time and (**b**) histogram of the frequency of each cycle time occurring in the data set.

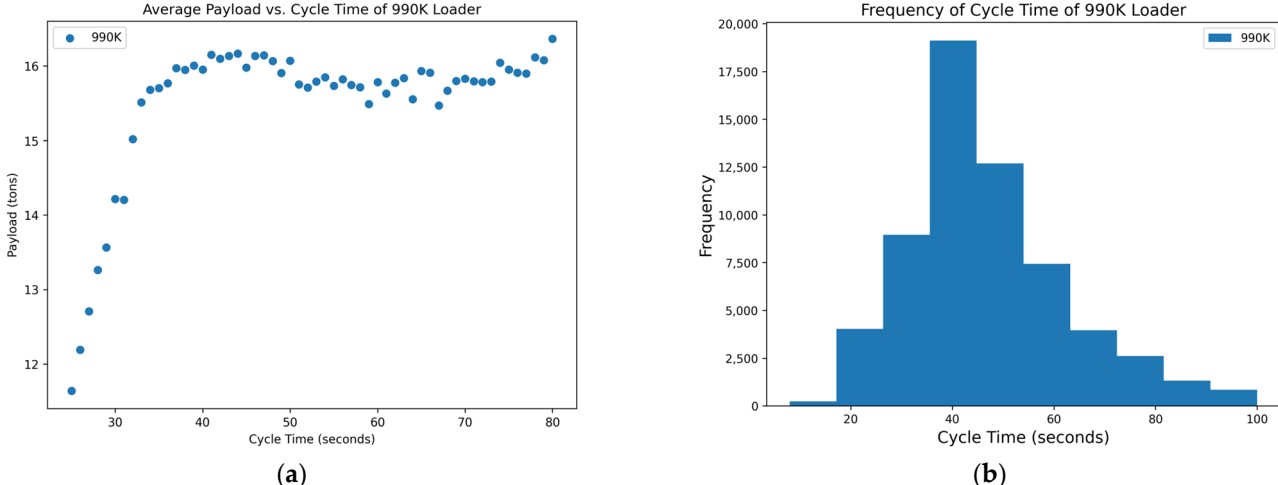

**Figure 2.** Distribution of variables within the 990K data set: (**a**) scatterplot of average payload per bucket and cycle time and (**b**) histogram of the frequency of each cycle time occurring in the data set.

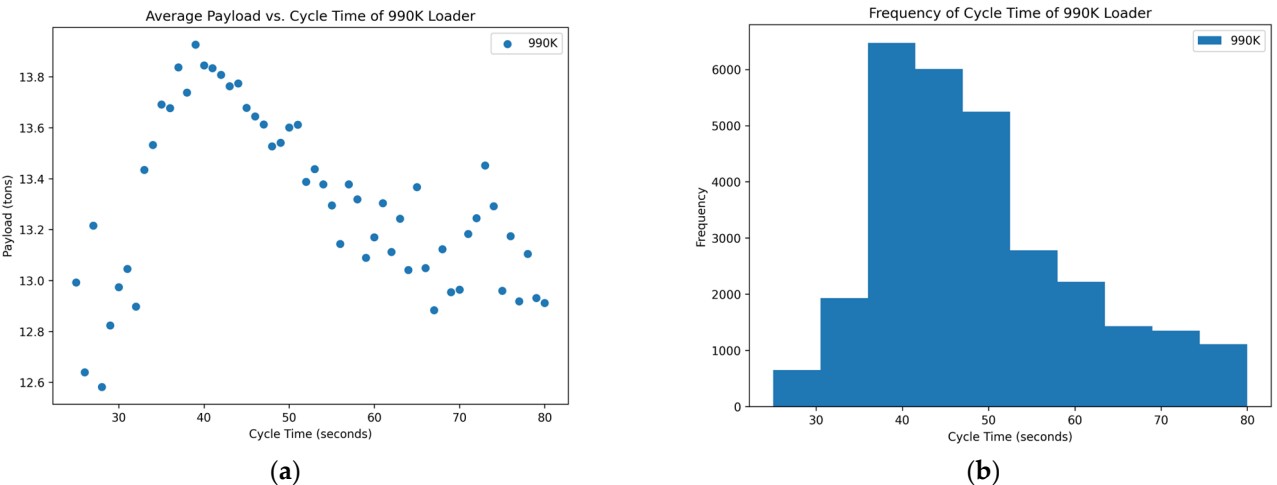

**Figure 3.** Distribution of variables within the 990K data set: (**a**) scatterplot of average payload per bucket and cycle time and (**b**) histogram of the frequency of each cycle time occurring in the data set.

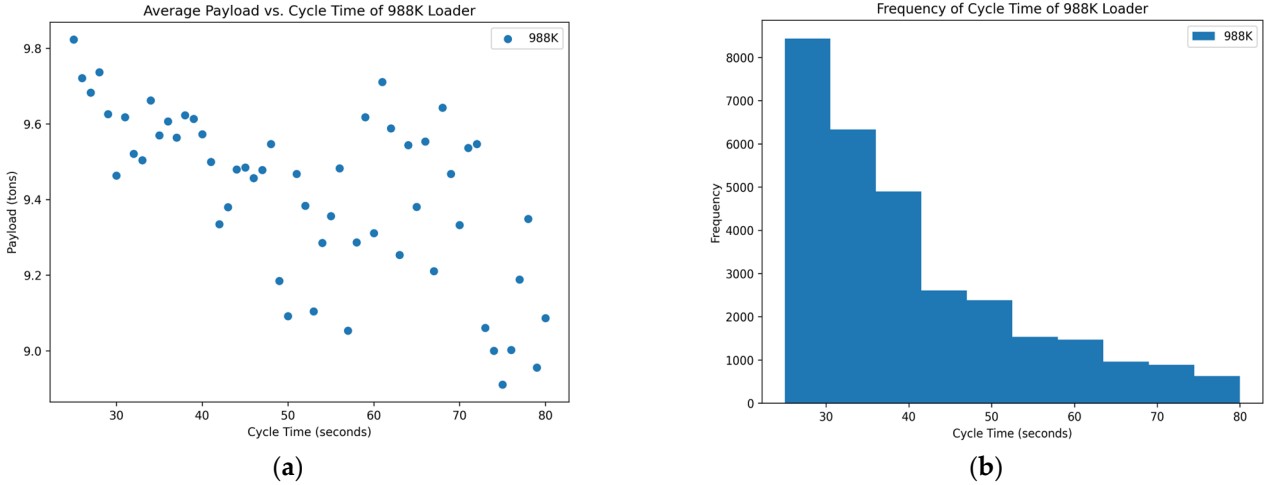

**Figure 4.** Distribution of variables within the 988K data set: (**a**) scatterplot of average payload per bucket and cycle time and (**b**) histogram of the frequency of each cycle time occurring in the data set.

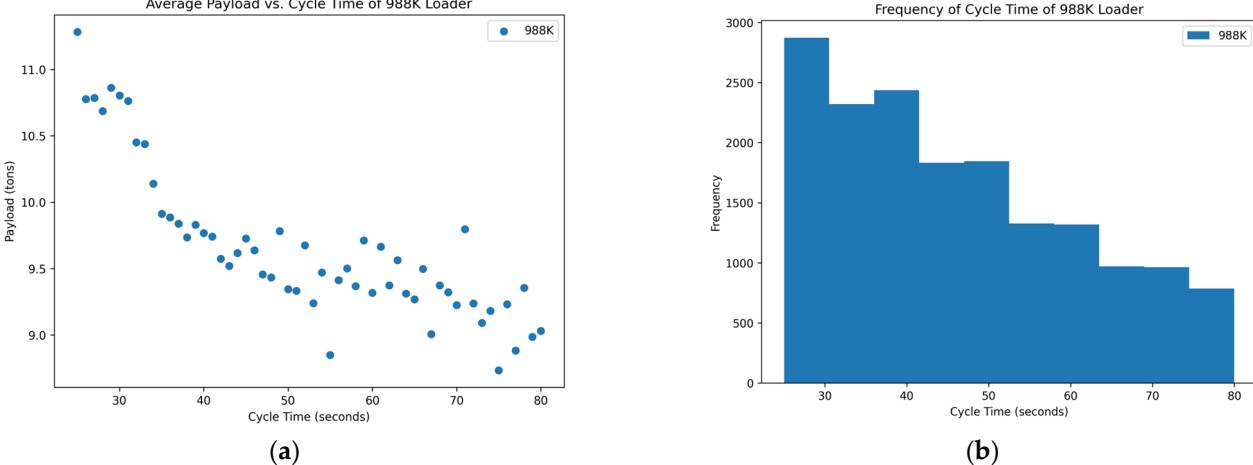

**Figure 5.** Distribution of variables within the 988K data set: (**a**) scatterplot of average payload per bucket and cycle time and (**b**) histogram of the frequency of each cycle time occurring in the data set.

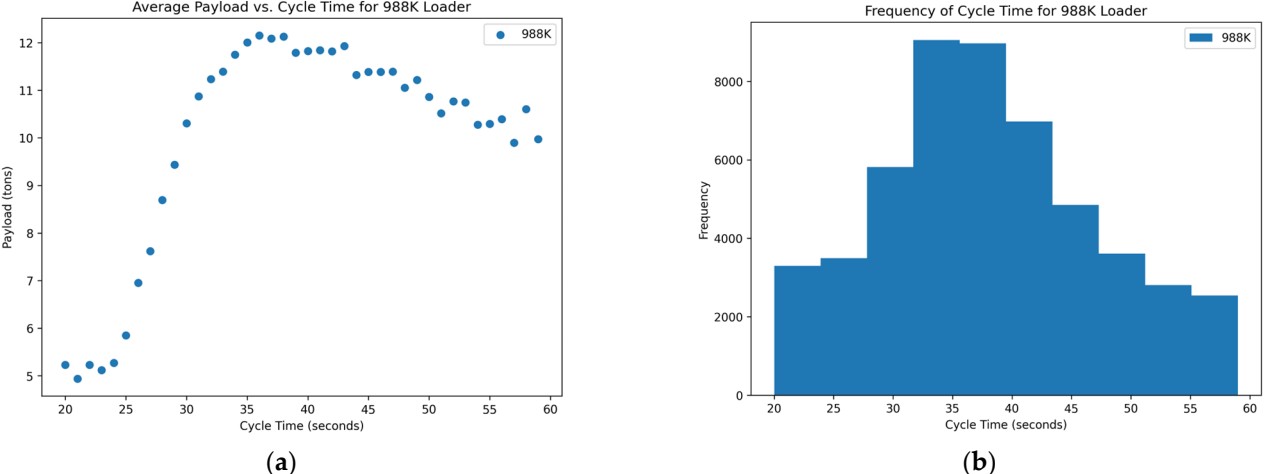

**Figure 6.** Distribution of variables within the 988K data set: (**a**) scatterplot of average payload per bucket and cycle time and (**b**) histogram of the frequency of each cycle time occurring in the data set.

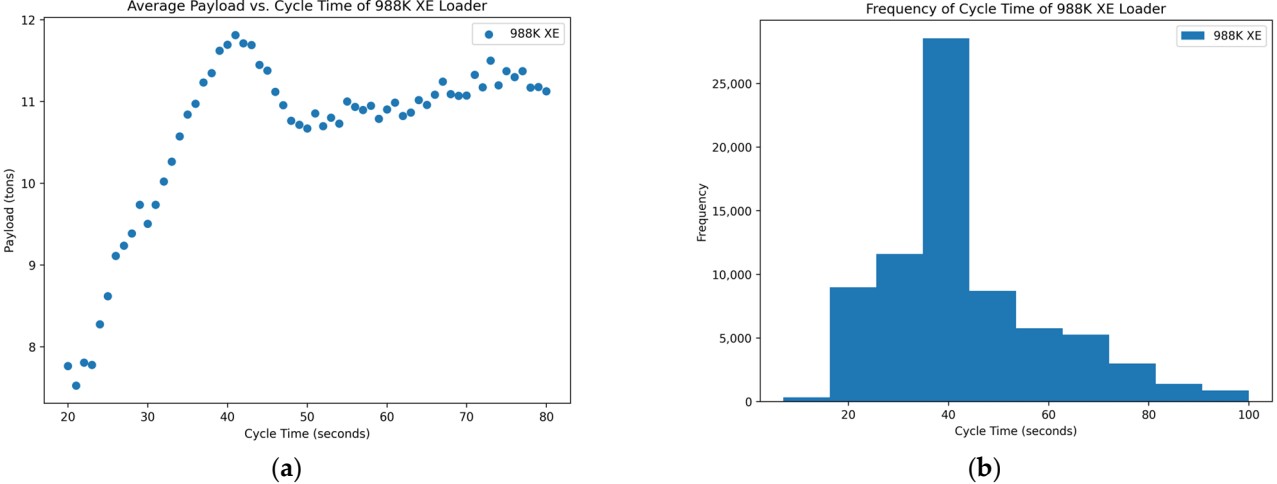

**Figure 7.** Distribution of variables within the 988K XE data set: (**a**) scatterplot of average payload per bucket and cycle time and (**b**) histogram of the frequency of each cycle time occurring in the data set.

Of the data that were collected and displayed for the seven loaders in the rock quarries, three of them showed promising results, two showed inconclusive and two showed poor results concerning the cycle times and yields. The scatterplots of Figures 3, 6 and 7 show the promising results. These figures indicate an optimum point around a cycle time of about 35–40 s in which the payload results in a higher yield. The trends of the scatterplots illustrate an increase in payload yield as they approach the higher bound of the cycle time. The increase in the payload is theorized as the fact that in a longer cycle time, operators repeatedly shovel into the blasted rock piles to gain higher fill factors for their buckets. The scatterplots of Figures 1 and 2 show the inconclusive results. Figures 1 and 2's plots are inconclusive because although there appears to be an optimum point for a cycle time that averages a high payload yield, it appears to level out after this point and only has slight changes to the average payload at each cycle time. The optimum point for these figures also appears to be around 35 to 40 s. Figures 4 and 5 show the poor results. Figures 4 and 5 have a negative trend with many outliers without a clear optimum point for high production except at the lower bound for the cycle time. The lower bound of the cycle time was positioned outside of generally accepted reasonable cycle times, which could suggest poor results.

The histograms of the frequency of each cycle time for each loader mainly show a normal distribution around the cycle times that represent the optimum point for a high yield in the payload. A few of the histograms, namely in Figures 2, 3 and 7, are skewed to the right, which could suggest the operators are multitasking rather than simply loading haul trucks. More data near the higher bound of the cycle time could suggest operators are taking unnecessary lengths of time to load a truck or clean the floor of the pit which could affect downstream production. The normal distribution in these graphs suggests that the operators of these loader units are performing exceptionally. This skew means that there are numerous more high payload data points at these cycle times which could inflate the data. Figures 4 and 5 show no normal distribution and instead are heavily skewed to the right. The right-hand skew goes against reasonable cycle times as they would be too fast at these points and would not reasonably be able to maintain a high payload yield and dump into the haul truck in the same minimum length of time. Figures 4 and 5 suggest either abnormalities or mistakes in the data and would need to be studied further with a site visit to analyze operations.

The two loader units from the surface coal mine were analyzed similarly to the seven loaders from the rock quarries. These loader units, however, did not have individual cycle times for each bucket payload in their data set. As a result, Figures 8 and 9 show the average payload per truck vs. the cycle time to fill that haul truck as well as the frequency of each cycle time shown next to it.

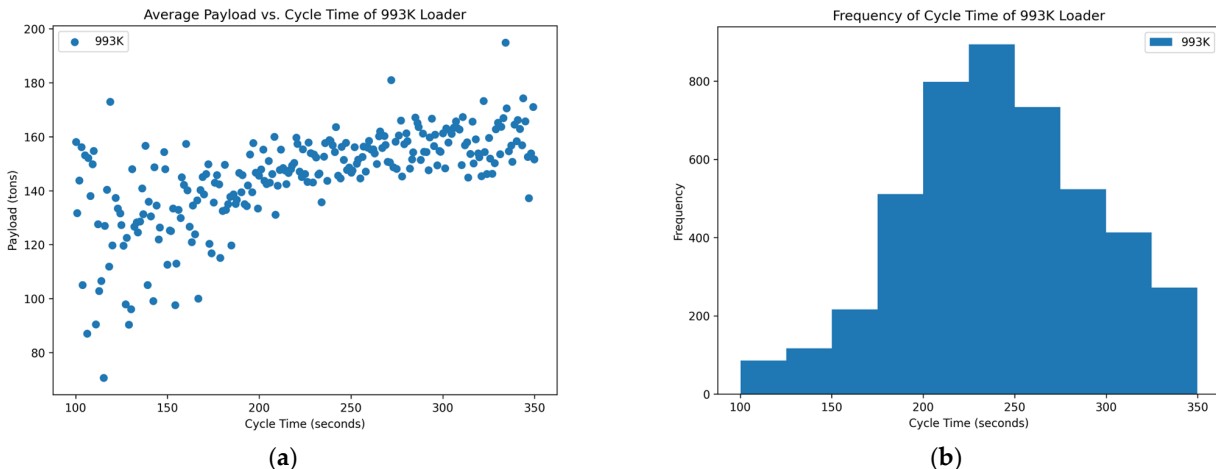

(a) (b)

**Figure 8.** Distribution of variables within the first 993K data set: (**a**) scatterplot of average payload per bucket and cycle time and (**b**) histogram of the frequency of each cycle time occurring in the data set.

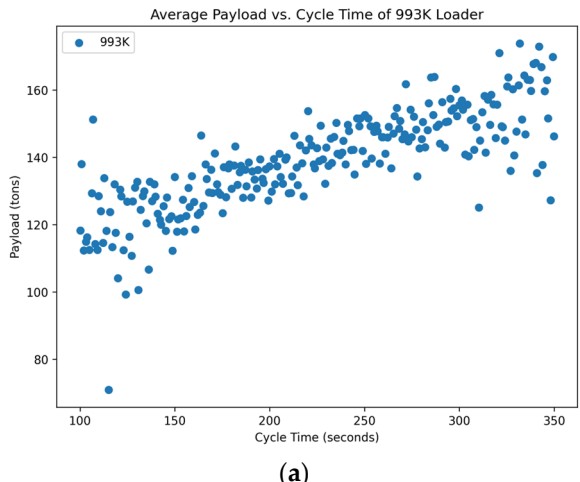
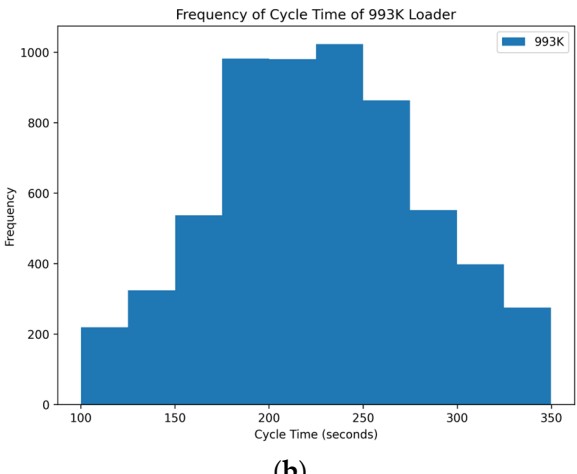

(**a**)  (**b**)

**Figure 9.** Distribution of variables within the second 993K data set: (**a**) scatterplot of average payload per bucket and cycle time and (**b**) histogram of the frequency of each cycle time occurring in the data set.

The two 993K loaders do not concretely contain an optimum point around a cycle time that shows high payload yields. Although the scatterplots do not show an optimum point, they do have a positive trend with some outliers. The positive trend confirms the hypothesis that as you increase the cycle time, you may have higher payload yields due to repeatedly shoveling into the blasted rock piles to achieve a higher fill factor. Although this is true, the longer loading strategy must be tempered because there could then be an unintended effect where loading one haul truck for larger amounts of time could back up the entire operation. The low production for the shift could be due to longer cycle times or a back-up due to long queue times at the loading site.

The histograms of the two 993K loaders at the surface coal mines primarily show a normal distribution centered around the cycle time of 250 s. This centered point could suggest that these larger operations, such as these surface coal mines, ideally operate around these times since they are more frequently hitting them. The normal distribution with a wide range also suggests that these loading units are not just used for loading haul trucks. They could be used for floor clean-up as well as moving material from stockpiles into crushers to continue production down the line.

*4.2. Statistical Analysis*

Statistical analysis was conducted on certain variables in the data to visualize their distribution for comparison. Cycle time, fill factor, payload, truck payload, and truck cycle time, were determined to be the best for determining the production performance of the loader units in their respective operation. The data were again split into two sets in this analysis based on quarry type. One data set included the rock quarries consisting of the seven loader units discussed in the data collection section. The other data set included the two loader units in the surface coal mines also discussed in the data collection set. The variables for the first set that were analyzed were the cycle time, fill factor, and payload (Figures 10–12). The variables presented in Figures 10–12 were chosen because all their data points were filled out across the seven loaders in the data set.

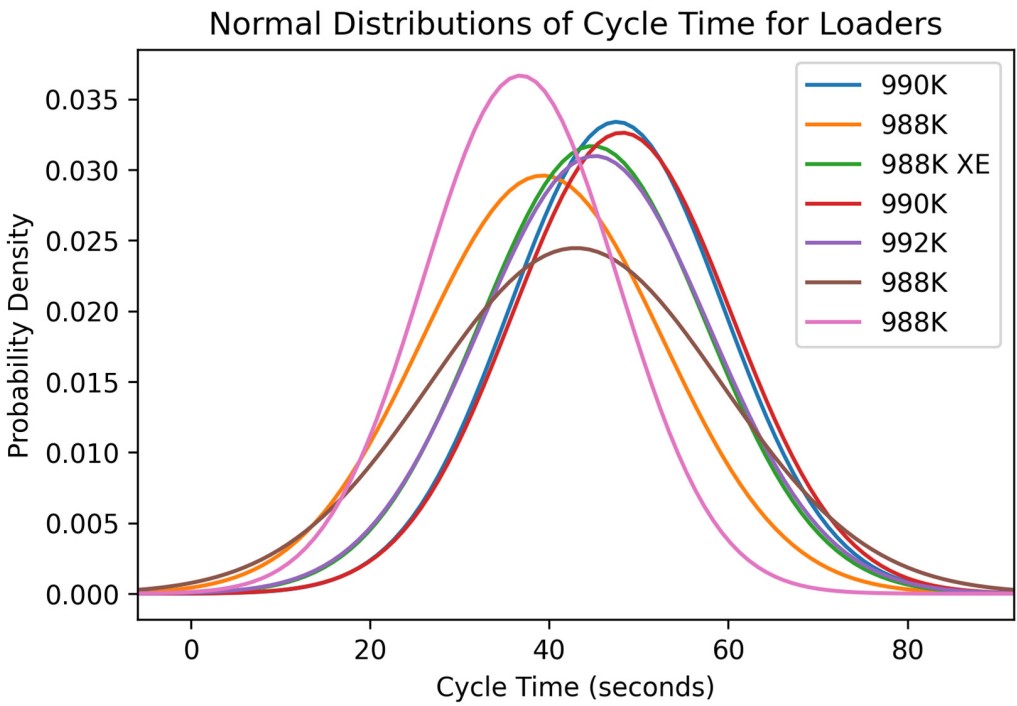

**Figure 10.** Normal distribution of the variable cycle time for the seven loaders in the rock quarries.

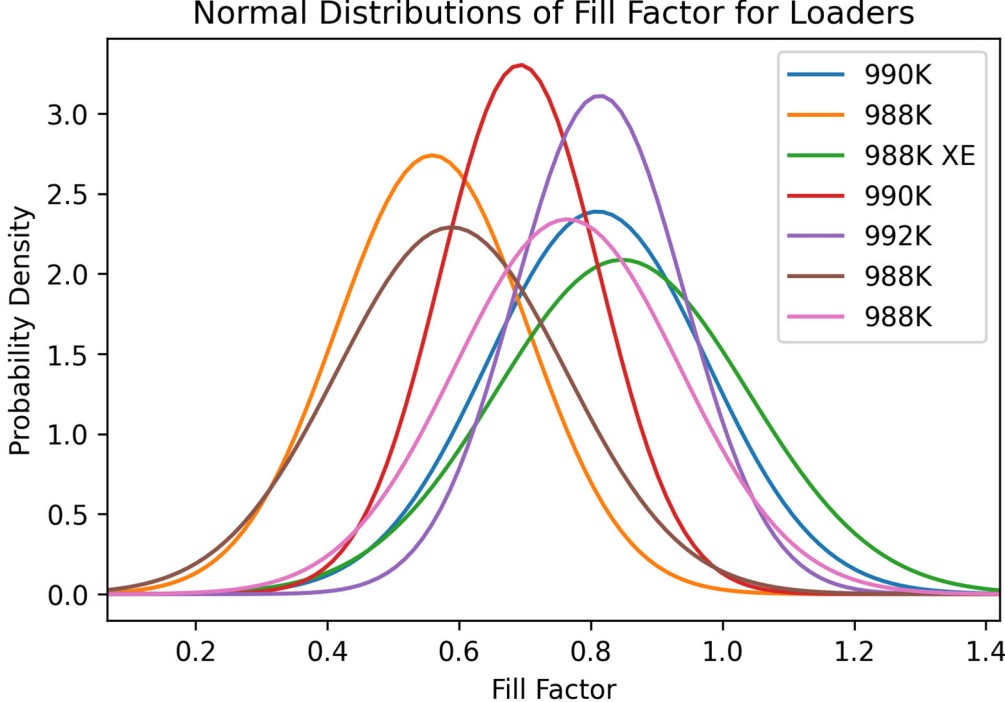

**Figure 11.** Normal distribution of the variable fill factor for the seven loaders in the rock quarries.

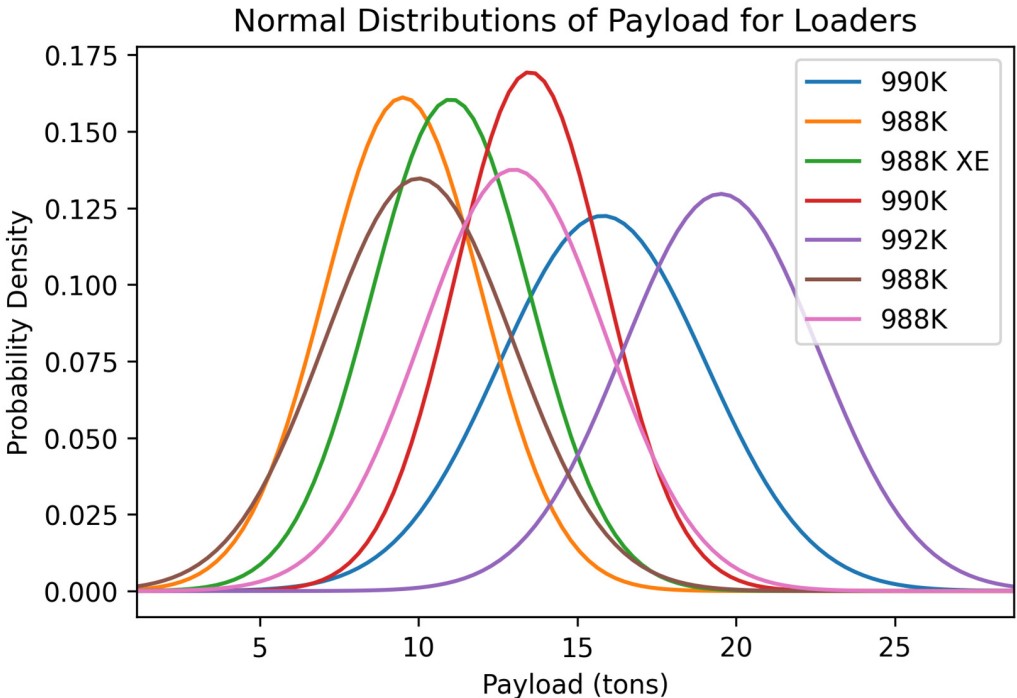

**Figure 12.** Normal distribution of the variable payload for the seven loaders in the rock quarries.

The normal distributions of the cycle times for the seven loaders were generally like each other. The distributions were centered around different values, but multiple distributions had similar size spreads. Multiple distributions were centered around a cycle time of approximately 50 s, suggesting that this is the average cycle time for rock quarries of this size. The two 988Ks had the best average cycle time of around 39 s, which indicated they were the fastest loaders, but looking at Figure 12 shows they did not have great payload values. The normal distributions of the fill factors for each of the loaders were considerably different. Many of them had low fill factors, indicating that the operators are not utilizing the size of their buckets to their fullest extent. One reason for the underutilization of buckets could be a lack of operator experience as well as loading units frequently being used for other purposes such as cleaning the face while registering these tasks as a loading cycle. Four loaders, 992K, one 990K, one 988K, and the 998K XE, showed optimal fill factors of about 80% or higher. The high fill factors are ideal because the operators are utilizing the machines to almost their full extent to fill the trucks, resulting in fewer cycles per truck and lower truck cycle times. The normal distributions for the bucket payload of each of the loading units vary considerably. Each generation of loading units has different sizes with different-sized buckets that allow for larger or smaller payload values. Although the distributions show their varying capacities, the quarries they operate in are nearly the same size as similar yearly production. Given the quarries' similar profiles and different machine options, the 992K loader would be the most beneficial for this type of operation with the correct operator. The 992K generally was one of the best-performing loading units in each of the variables shown in these distributions. The 992K had a higher average cycle time than some of the other loaders but remained near the lower range of the ideal cycle time for loaders in this type of operation.

Total truck payload, loader cycle time, and truck cycle time were analyzed in the second data set of surface coal mines consisting of the two loader units. The surface coal mine data sets for the 993K truck had more data filled out, allowing for the variables most related to production metrics to be analyzed. Figures 13–15 show the normal distributions of these two loaders. The variable bucket payload was not available for this data set because the software Cat MineStar Edge shows the total payload for each truck instead of each cycle's payload.

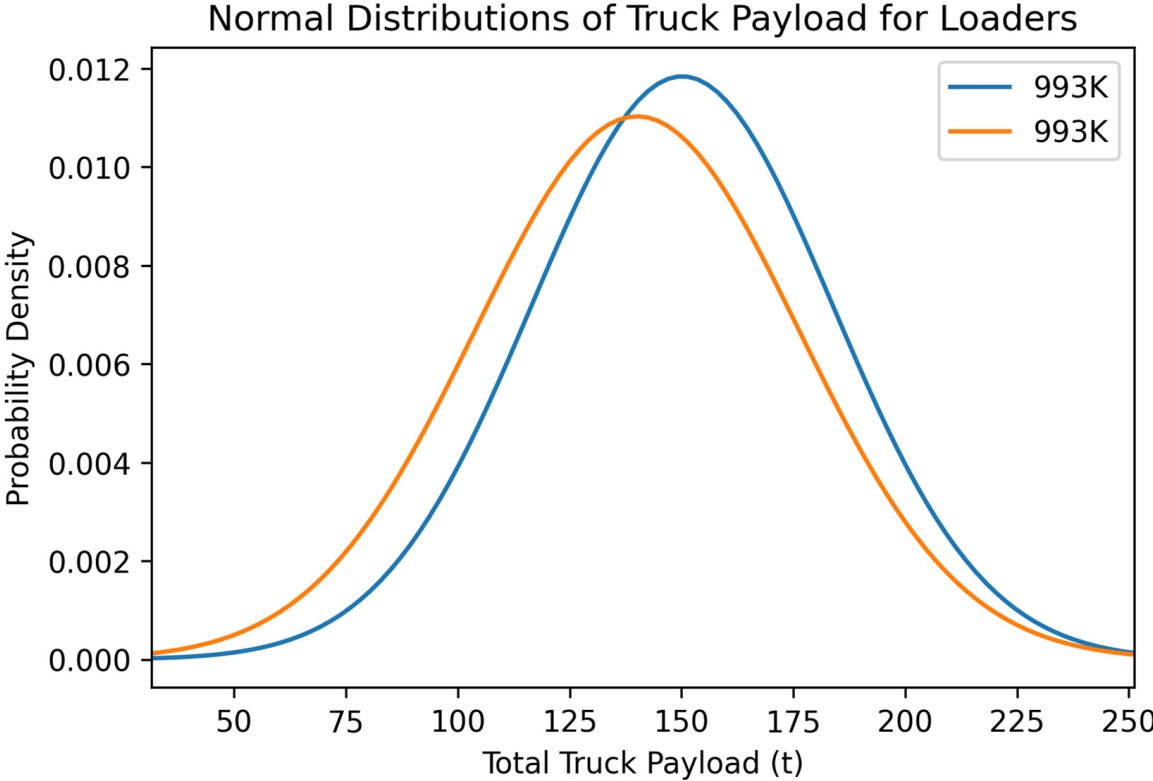

**Figure 13.** Normal distribution of the variable total truck payload for each loader in the surface coal mine.

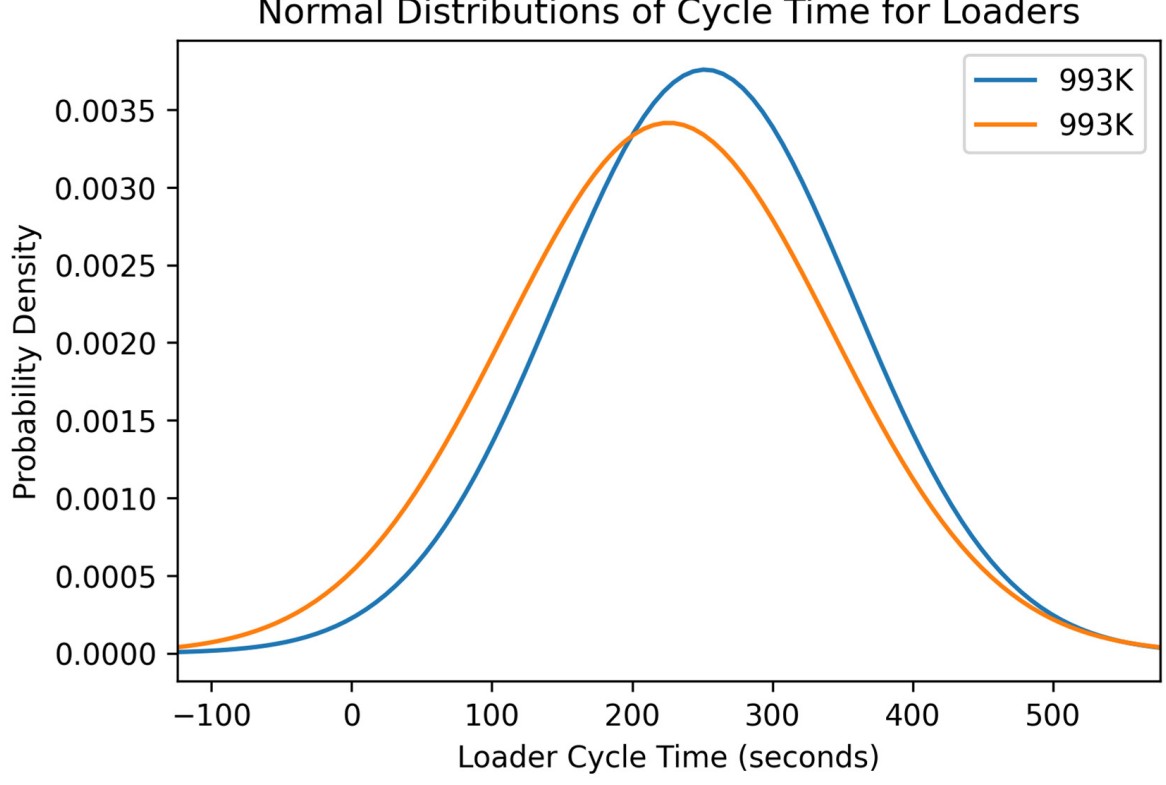

**Figure 14.** Normal distribution of the variable loader cycle time for each loader in the surface coal mine.

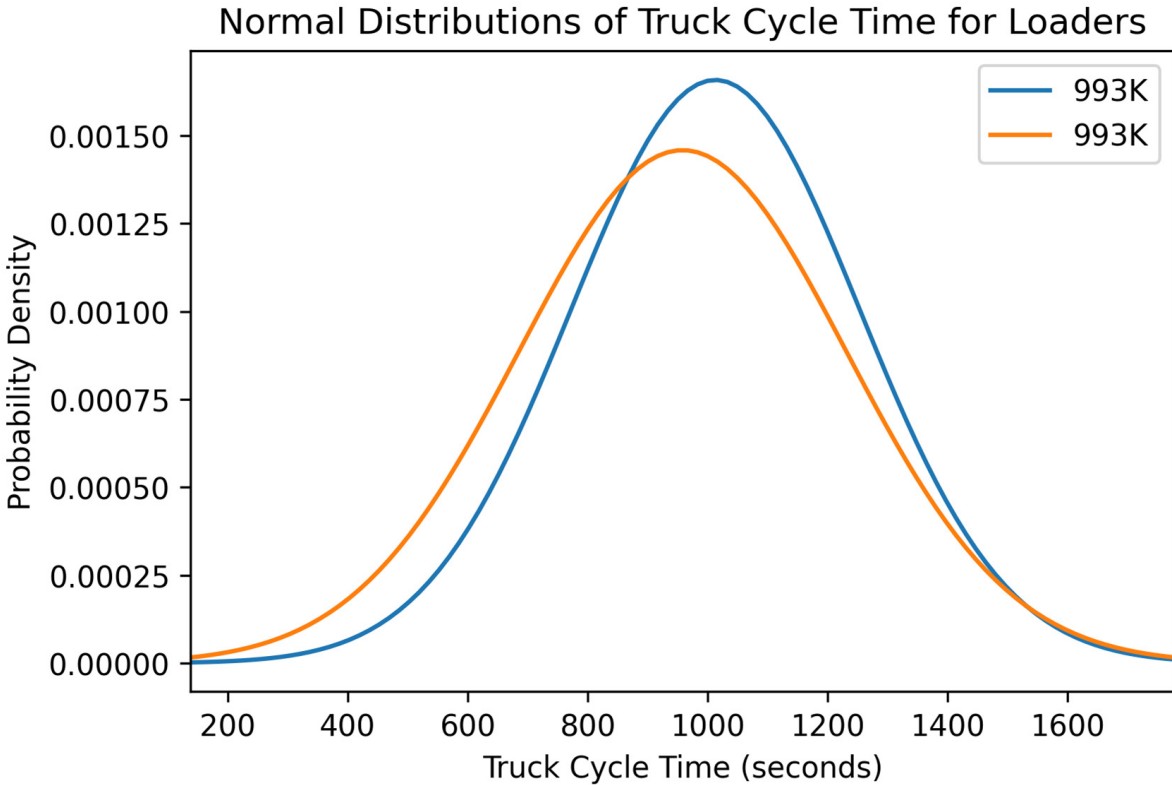

**Figure 15.** Normal distribution of the variable truck cycle time for each loader in the surface coal mine.

The two 993K loading units are the same type and size of machine, but one of the units outperformed the other in two of the three variable distributions. Examining the total truck payload distributions in Figure 13, the 993K machine, represented by the blue line, had a higher total truck payload of approximately 15 tons. The 993K, depicted as the orange line in Figures 14 and 15, had lower loader and truck cycle times by a small margin. In the future, the 993K loader could show higher production even though it lagged slightly behind in cycle time. Production variables can also be heavily influenced by operator experience, the location of operation within the same mine, and the type of material the loading units are working on within the quarry. Different materials could be harder to dig into or fully fragmented from blasting, allowing for easier digging and impacting production.

*4.3. GET Maintenance Applications*

Using the payload production values from Cat Productivity and the maintenance records of GET changes, an analysis was conducted on the change in production before and after maintenance. Only one maintenance record was obtained for a single 988K loading unit from one rock quarry. This maintenance record contained only the dates of maintenance. This was paired with production metrics throughout the time of maintenance for the purpose of analysis. Current maintenance practices are determined by the machine operator. If they believe that the teeth are worn out based on the feel of the machine, then they request a change. Unfortunately, many operators may not record when they perform maintenance or use paper records, which can easily be misplaced, to identify when GET maintenance operations have occurred. Figure 16 shows the average production of each selection of days leading up to maintenance. The selection of days leading up to maintenance means that $-4$ identifies the four days preceding maintenance, $-2$ identifies the two days preceding maintenance, 2 identifies the two days after maintenance, and so forth up to five days before and after maintenance. Figure 17 shows the percentage change in production value. The maintenance date is based on the date of each change in GET conducted on the 998K machine. The average payload used in the production cost analysis

was determined by taking the set of days from the previous maintenance day up to the day in question and then also taking the set of days from the next maintenance day to the previous day in question. The percentage change was calculated based on these days.

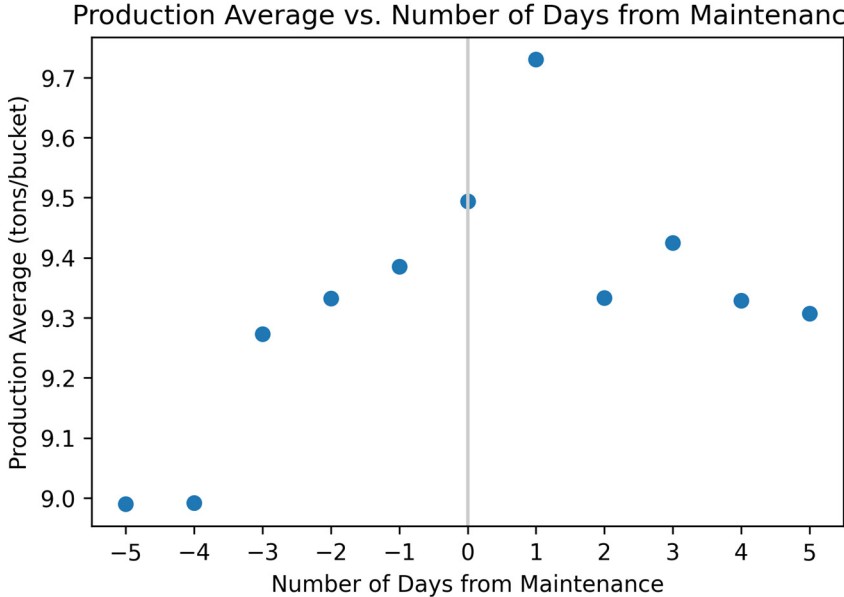

**Figure 16.** For one of the 988K loaders, the average production for each set of days leading up to and after maintenance on the GET.

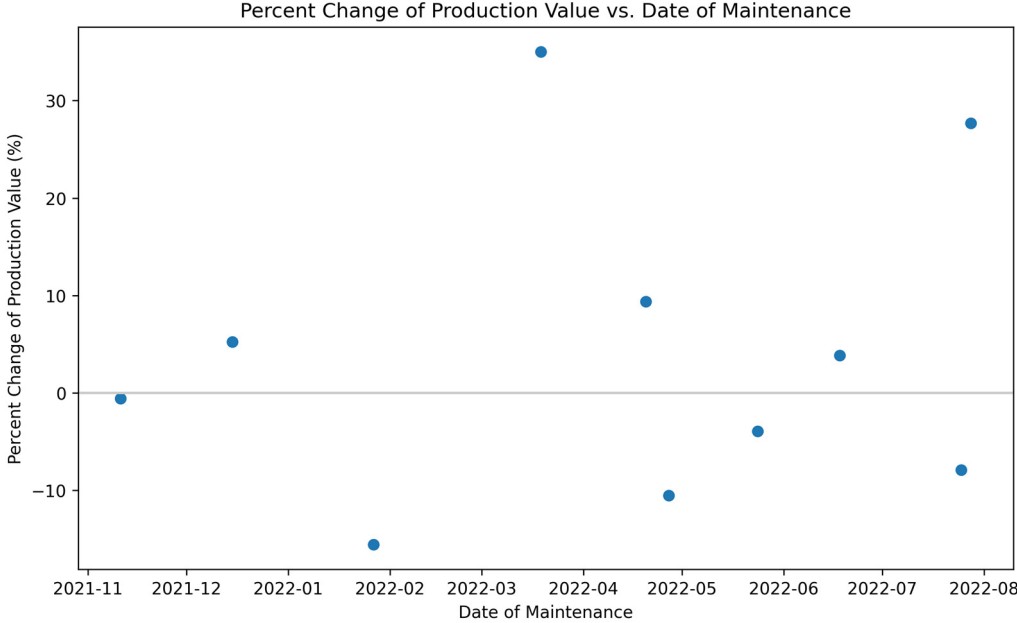

**Figure 17.** For one of the 988K loaders, the percentage change in production value from before and after the date of maintenance.

The production average leading up to and after maintenance presented in Figure 16 suggests there is no distinguishable material change in production once the GET were replaced on the loader bucket. Two sets of production days, −4 and −5, have low average payloads, but when comparing the sets of −3 to −1 to the sets of 2 to 5, there is relatively no change in the average production. Production is increasing before maintenance, and on the day of maintenance, it is high compared to the other values. The high value could be due to maintenance being performed at the start of the shift but there is no other indicator

besides the date in the records of the time when maintenance occurred. The set of one day after maintenance has the highest production average but has the potential to be considered an outlier when tied with the sets of days after maintenance.

Figure 17 illustrates the percentage change in production value around the day of maintenance and suggests there is no distinguishable material change in production due to maintenance. The percentage change spans an even split of values in negative and positive change. It is worth noting that the positive percentage change has a higher range going up to approximately 32% while the negative percentage change goes down to only about −15%. Initially, it was speculated that if the GET were changed on the bucket, it would benefit the operation, and production would increase. This would be a result of the new teeth allowing for easier digging. Based on the data and the results calculated from these data, the results are inconclusive as to whether the current process of GET maintenance is beneficial or if operators are changing the GET too frequently. GET maintenance is generally determined by the operator's decision because they experience a harder time digging into the rock face or they visually notice too much wear. Therefore, for better results in the future, there should be a standard low measurement for the teeth that objectively indicates maintenance to be performed to change them out.

*4.4. Machine Learning*

Using Python, machine learning methods were used to analyze the data sets to initially determine correlations between the variables included. The first machine learning analysis used was to test linear regression. Because payload is a key indicator of performance, it was correlated to the cycle time. Using the associated linear regression libraries within Python, a test and a training data set were identified. The two sets were then used in the linear regression algorithm to test the variables. The correlation coefficients were output for the test set after training was complete. The correlations were all less than 0.01 for each loader, indicating that there was no significant linear correlation between the input variables from the loader and hauler machines.

Due to the poor results obtained from the linear regression model, the data were analyzed using a polynomial regression model. After inputting the data for all of the loaders and running them through the respective Sklearn polynomial regression library, Figure 18 shows a heat map of the correlation between the variables in the data set.

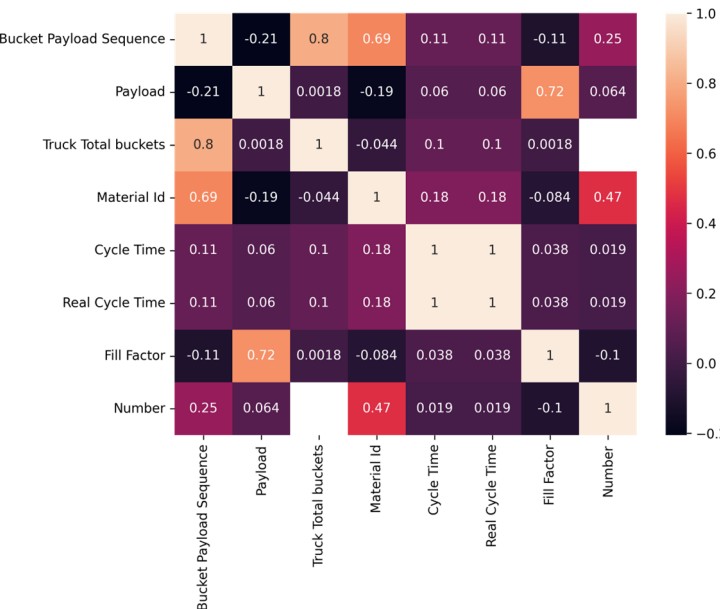

**Figure 18.** Heat map created using the polynomial regression machine learning algorithm to show the correlation coefficient between the variables in the loader's data set.

The polynomial regression algorithm results were not promising. Despite a higher correlation between multiple variables compared to the linear regression algorithm, a certain group of variables with a significant correlation between them did not emerge. This result was unexpected because the large amount of data in the data set was expected to at least suggest some form of correlation between a few of the variables. Namely, an expectation of a higher correlation between the variables of payload and cycle time would be anticipated. The correlation result between the payload and cycle time was only 0.06. We hypothesized that there would be a significant correlation shown, but with the data and the models created, this relationship was not found. Due to the unusual nature of the results, future research would need to be conducted with additional types of machines and more data with all variables filled out. This research would be beneficial for future model generation.

The next machine learning technique attempted was prediction models. The models utilized in this study were K-nearest neighbors, decision trees, random forests, and polynomial regression. The full data set was run through each of these machine learning classifier algorithms and output an accuracy score. This accuracy score identifies how well the algorithm can predict values based on the training data set initially used in the algorithm. Table 4 shows the accuracy scores of the four models from the greatest to the least.

**Table 4.** Accuracy scores of the four machine learning algorithms used on the data set for predictions.

| Machine Learning Method | Accuracy of Prediction |
| --- | --- |
| Decision Tree Classifier | 0.8799 |
| Random Forest | 0.8149 |
| K-Nearest Neighbors | 0.766 |
| Polynomial Regression | 0.455 |

Of the four algorithms used, the decision tree classifier was the most accurate followed closely by the random forest and K-nearest neighbors algorithms. With 87.99% accuracy, the decision tree classified model was able to estimate the performance of any loader based on the full analysis of the performance metrics input. Future research and additional loaders' data collection would allow for predicting a loader's production metrics in an operation given the type of operation and the type of loader.

## 5. Conclusions and Future Work

This exploratory study of data from nine loading units utilized data analysis, statistical analysis, and machine learning techniques. Seven of these loading units came from the software CAT Productivity while the other two came from the software CAT MineStar Edge. Key production parameters were identified using multiple variables which include bucket payload, loader cycle time per bucket, fill factor, truck cycle time, truck total payload, and loader cycle time per truck. Following these analyses, an analysis of production before and after maintenance of the ground engaging tools was conducted to determine the effectiveness of these maintenance practices. Data analysis was used to find a relationship between the average payload of each loader and their cycle times. This relationship was inconclusive in determining the optimum point at which production can be increased down the line. A large variety of data across each loader between cycle time and average payload was found. However, three loaders indicated an optimum cycle time between 35–40 s to yield a higher payload.

The statistical analysis conducted was used to compare the distribution of production variables across the loaders performing in a similar work environment. The Caterpillar 992K and one of the 990K loading units were found to be the highest-yielding machines. In the CAT MineStar Edge data set, one of the Caterpillar 993K loading units outperformed a similar 993K machine in all production variables for unknown reasons that will be the basis of future research. The influence on these results could be due to operator experience, the location of operation within the same mine, and the type of material.

Machine learning was initially used to investigate the correlations between the variables in the data sets. Through the linear and polynomial regression models that the data were put through, there was no significant correlation between any of the variables. Although there is a correlation between the plan distance full and the truck cycle time, this relationship is expected because the further the trucks travel then the higher the cycle time. This indicates that future research should be carried out to investigate why this correlation was not what it theoretically should be. With the use of four prediction algorithms, the decision tree classifier algorithm produced the best results in estimating the performance of a loader based on a full analysis of all the data with an accuracy of 87.99%. The next best model was random forest with an accuracy of 81.49%. In future operations, given the type of loader, we will be able to predict what its production metrics should be.

After generating an average production comparison before and after maintenance and a percentage change in production value for the ground engaging tools on the loader bucket of one of the 988Ks, it was found that there was no material change in the average production of the mine. This analysis still does not answer the question of whether the GET are replaced too frequently. A future analysis would attempt to create a prediction model for optimal maintenance intervals for the GET.

The CAT Productivity data varied greatly from CAT MineStar Edge as they were found to be less detailed and largely subject to human error since numerous variables needed to be deleted. The loading task is repetitive because the cycle is short (less than a minute each) and performed over long shifts (eight to ten hours). Data reliability is low due to operator input. Future work could help improve the models described in the paper. An additional year of research will provide more data and yield more precise models. Additional digitization of machine information will populate the software with additional variables and a wider variety of machine types for further refinement of predictive models. This would also include data from haul truck units as well. As remote monitoring popularity increases, more companies will pay for subscriptions to this software, allowing for more machines to be used in analyses. Site visits would help with future research to better understand the operation and everyday tasks performed by each loader. Irregularities in the data could be identified by not only these site visits but also time studies could be included to determine choke points in operations. Additional research should be carried out on better ways to determine whether GET are worn out instead of just operator opinion deciding when they should be changed, as this can eliminate the potential for human error.

**Author Contributions:** Conceptualization, B.G.; Methodology, B.G. and B.N.; Software, B.G.; Investigation, B.G.; Writing—original draft, B.G.; Writing—review & editing, B.N. All authors have read and agreed to the published version of the manuscript.

**Funding:** This research received no external funding.

**Data Availability Statement:** The quarry-specific data are not publicly available due to private identifying information of companies used in the study.

**Conflicts of Interest:** The authors declare no conflict of interest.

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
