# Peer review of "Data Analyses of Quarry Operations and Maintenance Schedules: A Production Optimization Study"

_mining, doi:10.3390/mining3020021_

Round 1

Reviewer 1 Report

The title of the article "Optimization of Quarry operation.." is not consistent with its content, because at no point in the article was optimization of quarry operation. The analysis of the effects of this optimization was not carried out so it’s hard to write about optimization. In the article, some statistical analyzes related to the work of loaders in mines were made, however, these analyzes are relatively shallow and do not meet the standards that should characterize statistical inference in a scientific article.

Below, in detail, I will point out the basic shortcomings of the article.

- Literature analysis is definitely superficial. Although a dozen or so literature citations were indicated, a critical assessment of the development of methods and science was not made and the connection with the research conducted in the article was not shown. Literature analysis is primarily a collection of generalities and often obvious statements. The analysis of the literature in the field of mining machinery maintenance, data acquisition methods, and machine methods used for mining machinery should definitely be enlarged. For example, the authors write in line 103 [Machine learning has been a popular topic.....] and cite only 4 articles, where the achievements of the world literature in the field of mining machine analysis are really impressive.

- The key drawback of the methods in this paper is that the statistical analysis of the input data and inference is incorrect. For example, it has not been shown that the technological parameters of the loaders are normally distributed (Fig. 10-15). There is no appropriate statistical test to confirm this, especially since previous analyzes show skewness of distributions. It is also not known what the purpose of determining these distributions is.

- Figures 10-15 are redundant. The average reader of the article knows the shape of the normal distribution curve. The parameters of these distributions were better described in a table, together with proof that it is a normal distribution.

- It is not clear how GET maintenance would affect work efficiency. There is no specific description of the factors of the excavator's work technology. The performance analysis is carried out for a very short period. Perhaps it should be concluded in the longer term, where the impact of using GET maintenance would be demonstrated. And perhaps there are too many factors that affect operation times and productivity that cannot be proven. The article lacks an in-depth analysis of this problem.

- It is not known what machine learning methods have shown. Were these methods necessary here and proved to be better than simple regression analysis? In what aspect? What has ML really proven from the point of view of machine technology in the mine? There is no clear indication of the relationship between the ML method used and mining production and how this production (performance) was in the model calculated?

- The article contains many unjustified and unsubstantiated statements. Examples:

349 More data near the higher bound of the cycle time could suggest operators are taking unnecessary lengths of time to load a truck or clean the floor of the pit which could affect downstream production.

How do you know that? Is there any proof? Which data from the analysis confirms this?

414 One reason for the underutilization of buckets could be a lack of operator experience as well as loading units frequently being used for other purposes such as cleaning the face while registering these tasks as a loading cycle.

How do you know that the operators were inexperienced? Is there any proof? Which data from the analysis confirms this?

425 Given the quarries' similar profiles and different machine options, the 992K loader would be the most beneficial for this type of operation with the correct operator.

The article did not analyze the operating costs or profitability of using this type of machinery, so where did this statement come from?

452 Production variables can also be heavily influenced by operator experience, location of operation within the same mine, and the type of material the loading units are working on within the quarry.

Has it been investigated? It is somewhat obvious that such a situation could take place, but how has it been proven?

I believe that the article needs to be thoroughly revised and such statements should be removed from the article.

Other minor imperfections noted:

- the description of the GET is missing, so it is not known how the device replacement time is determined and how the decision-making process is organised.

- no differences in the use of GET were shown, so the article did not show that optimization can be applied through the use of additional analyses

- the description of the work in Excel and other tools is too accurate (3.3. Data Analysis) - these are unnecessary details and look more like a research report than a scientific article, especially since the tools used are rather basic.

Reviewer 2 Report

Firstly, the paper could benefit from providing more information about the data collection process, such as the specific sensors used to collect the real-time data from loaders and haul trucks. Additionally, more information on the type and size of the quarries used in the study could provide a better understanding of the study's scope and applicability.

Secondly, the paper could provide more details on the Decision Tree Classifier algorithm used and how it was trained. A more detailed description of the algorithm could help readers better understand the model's accuracy and limitations.

Lastly, the paper could benefit from providing more context on the implications of the study's findings. How can mining companies utilize these findings to improve their operations? What are the potential challenges of implementing the proposed preventative maintenance schedules? More information on the practical applications of the study's findings could provide valuable insights for mining companies looking to optimize their operations.

See general comments in the attached document.

Reviewer 3 Report

Please go at the suggested recommendations in the following:

1.      I recommend rewriting the sentence in order to make it stronger and to provide an accurate amount or percentage of the final result. Page 1

2.      Authors should mention the various sections and include the paper's structure.

3.       rewrite to be clear which results? page 3 lines 110-115.

4.      Description of Heuristic Algorithm in this section you mentioned the constraint without any details. Page 6, lines 276-…..

5.      Give us more explanation for this figure. Page 12, figure 10  

6.      Please check this paragraph to be clear with specific results of the benefits for this study, may be need support with equations. Page 17, lines 501

7.      The conclusion should be rewritten by giving the main point summarized.  specify the standard or establish comparison with others. Page 18, lines 556-……

Regards

Round 2

Reviewer 1 Report

The authors clarified my doubts. Not all elements of the article have been corrected in a proper way (e.g. fig. 11-15 do not contribute much), however, I believe that the article in this version is suitable for publication.